# Testing the Limits of the Spatial Approach:
# Comparing Retrieval and Revisitation Performance of Spatial and Paged Data Organizations for Large Item Sets

Carl Gutwin[†], Michael Van Der Kamp[†], Jeremy Storring[†], Andy Cockburn[‡], and Cody Phillips[†]

University of Saskatchewan

University of Canterbury

## ABSTRACT

Finding and revisiting objects in visual content collections is common in many analytics tasks. For large collections, filters are often used to reduce the number of items shown, but many systems generate a new ordering of the items for every filter update – and these changes make it difficult for users to remember the locations of important items. An alternative is to show the entire dataset in a spatially-stable layout, and show filter results with highlighting. The spatial approach has been shown to work well with small datasets, but little is known about how spatial memory scales to tasks with hundreds of items. To investigate the scalability of spatial presentations, we carried out a study comparing finding and re-finding performance with two data organizations: pages of items that re-generate item ordering with each filter change, and a spatially-stable organization that presents all 700 items at once. We found that although overall times were similar, the spatial interface was faster for revisitation, and participants used fewer filters than in the paged interface as they gained familiarity with the data. Our results add to previous work by showing that spatial interfaces can work well with datasets of hundreds of items, and that they better support a transition to fast revisitation using spatial memory.

**Keywords**: Filtering, spatial memory, revisitation.

**Index Terms**: H.5.m. Information interfaces and presentation (e.g., HCI)

## 1 INTRODUCTION

Visual presentations of large datasets – such as a visual display of a photo collection or a 3D asset catalog – are now ubiquitous, but finding and revisiting items in these workspaces can be difficult. Many systems therefore provide filters that reduce the size of the displayed dataset, with items displayed in a set of pages that the user can browse through. However, presenting results in a series of dynamically generated pages presents a usability problem when users need to return to items they have looked at before. When filters are applied, the pages and items within them are typically reorganized, so previously viewed items will not be shown where they used to be. Every time a user wants to revisit an item, they must recall and reapply the relevant filters; this can be a laborious and frustrating process, particularly when users go back to an item multiple times.

first.last@usask.ca
andrew.cockburn@canterbury.ac.nz

To ease this problem, many websites incorporate features to help users revisit items. "Recently viewed" lists, temporary baskets to collect items, and separate comparison pages are now common. Researchers have also proposed other methods such as read wear, which adds visual marks to objects that have been inspected [3,18]. These strategies can be effective, but often fail to fully support the user's revisitation needs. For example, if the user's working set is larger than the recency cache, the needed item will not appear in the "recently viewed" display; users also often ignore comparison baskets because they do not know while looking at an item that they will want to return to it later.

### 1.1 A Potential Solution: the Spatially-Stable Approach

A solution to the problem of re-finding with filters is to show all items on the screen at once, and show filters through highlighting rather than in new pages. Prior studies with command icons [16,17,29,30] and document navigation [9] have shown that spatially stable interfaces allow faster item revisitation as users gain experience with the layout. A spatial layout means that if an item of interest is in the top-right corner when first laid out, it will always be in that location – allowing users to build spatial memory of where things are. Once locations are learned, users can quickly retrieve items without needing to visually search for them.

This potential benefit, however, comes at a cost: showing all items at once means that when the collection is large, each item's representation must be small, allowing less detail; in addition, the presentation of an entire large collection could seem overwhelming for the user (Figure 1). Previous studies have only tested the spatial approach in small datasets, and although human spatial memory has a large capacity, there is little known about whether the approach's advantages will occur with large sets of items.

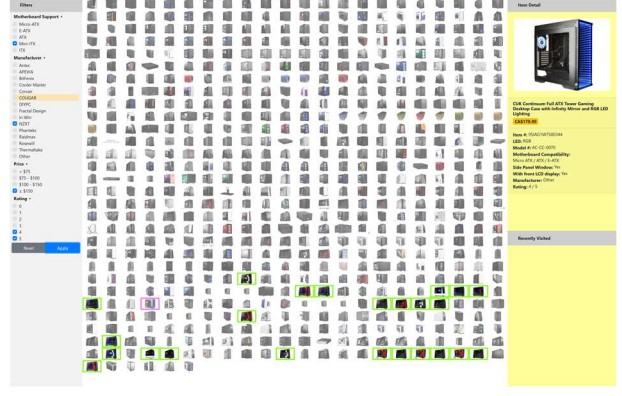

Figure 1: Spatial interface showing 700 graphics cards from a product catalogue (filter panel at left, detail panel right). Green highlight shows matching items, with non-matching items dimmed; pink box shows currently-selected item.

Graphics Interface Conference 2020
28-29 May

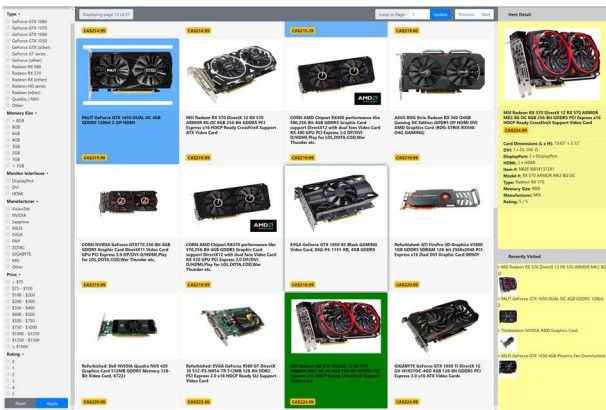

Figure 2: Paged interface showing the same collection (filters panel at left, detail panel right). Blue highlight shows items inspected in this trial; green shows current selection.

We carried out an evaluation to determine whether spatially-stable interfaces perform well with larger collections than what has been tested previously. Our spatial interface showed all 700 items on the screen at once, with filtering shown with highlighting and dimming (Figure 1). Our comparison interface used a standard paged presentation in which the current filter subset was presented in a set of dynamically-generated pages (Figure 2). Participants were asked to find and revisit six targets over eight blocks of trials, with different amounts of cue information to simulate different levels of recall about item details. Our primary measures were retrieval time, the number of filter actions carried out for each trial, perceived effort, and preference.

The study provided several new results:

- The spatial interface was slower for initial finding, but was significantly faster for revisiting previously-seen items; overall completion times were similar.
- Filter use differed substantially: the paged interface was highly dependent on filters across all blocks, whereas filter use consistently decreased for the spatial interface.
- Effort ratings were very similar for the two interfaces.
- Preferences were divided: participants stated that the spatial interface was faster and better for remembering, but 13 of 20 preferred the paged interface overall.

The study showed that spatially-stable presentations can work well for datasets of several hundred items – but also showed the limitations of the approach (finding items for the first time, and participant preference). Our results provide guidance for designers who wish to improve revisitation support in large visual workspaces, and add to knowledge about the strengths and limits of the spatial approach.

## 2 RELATED WORK AND ITS IMPLICATIONS FOR DESIGN

Visual workspaces are a common setting for exploration, browsing, or analysis of visual items and documents. These kinds of tasks often involve open-ended and ongoing information seeking [40], and depend more on actions like selection, navigation, and "trial-and-error tactics" than on the formulation of precise queries [26].

Within these visual spaces, Shneiderman's 'visual information seeking mantra' [34] provides a useful starting point for the design of systems that aim to support users in finding, exploration, and navigation. Derived from the design and evaluation of a wide range of visual systems (such as the FilmFinder's 'Starfield' visualization [2]), the mantra notes that users are often best supported when the system provides an *overview*, allows users to *zoom and filter*, and then to request *details on demand.* Although Shneiderman's mantra advocates initiating navigation with an overview, the size and type

of the dataset will influence the viability of doing so. Providing an overview with a discrete representation of thousands of data items is infeasible due to clutter; however, many task domains involve fewer items, and a spatially stable overview is viable. In terms of Shneiderman's mantra, our design approach supports *overview, ~~zoom and~~ filter, details on demand.* Zooming is intentionally excluded from our design because it impairs spatial stability. Instead of zooming, our design uses selective item highlighting when filters are applied.

### 2.1 Organizing Data in Visual Workspaces

HCI researchers have long been interested in how different ways of organizing data affect users (e.g., since early studies of how people organize their desks [25] and studies of deep and broad menu structures [22]). Spatial organization of data is common in real-world settings, and HCI researchers have investigated this approach for digital data in several contexts. For example, Shipman and colleagues investigated "spatial hypertext," which shows relationships among documents through relative location, and uses spatial stability to help people remember documents [34]. Robertson and colleagues also used spatial stability in the design of the Data Mountain [28], a technique that allowed users to arrange bookmarks on a receding 3D plane; studies showed that the spatial layout allowed faster retrieval, even after a three-month gap.

Exploratory-search interfaces have also used spatial organization. Several systems cluster results into categories (e.g., [6]) or into hierarchical facets (e.g., [18]) which uses spatial layout to indicate structure and to support analysis and interactive search. Studies have shown that clustering can improve browsing of large collections (e.g., [6]). Studies have also investigated visual search strategies when users explore spatially-arranged collections (e.g., showing that visual search is influenced by landmarks and previously-visited locations) [21]. Spatial approaches are also common in information visualizations such as scatterplots [2]; however, this avenue of research has not compared spatial techniques to other types of organization (such as paging).

### 2.2 Techniques for Finding Things in Visual Spaces

A review of interface techniques for finding items in large visual information spaces identified four broad categories of approaches [8]. *Overview+detail* interfaces provide a spatial separation between an overview of the information space and a detailed view of a particular item or region. *Zooming* interfaces use a temporal separation between subsets at different magnification levels. *Focus+context* interfaces seek to minimize the seam between overview and detailed (zoomed) views by blending a focal region within its surrounding context. Finally, *highlighting* techniques rely on visual embellishments to bring forward or suppress items in the display. The spatial interface discussed in this paper combines overview+detail and highlight-based techniques.

A few researchers have also carried out studies involving spatial and paged data organizations. For example, Kim and colleagues [23] compared performance of scrolling and paged presentations of data, and found that paging led to faster selection and higher likelihood of finding relevant items. Scarr and colleagues compared a spatially-stable grid of icons that scaled the grid to account for changes in window size, to a window that re-flowed the icons to fit the window shape; they found that the spatial approach was faster than re-flow and was resilient to resizing [32].

### 2.3 Information Revisitation

People's patterns of behavior in many domains are known to be highly skewed, both in terms of recency and frequency of use. In natural language, word frequency is inversely proportional to the word's rank in a frequency table [1]. In HCI, related observations have been made regarding the frequency of commands [12], the

distribution of visits to webpage URLs [37], and the use of e-mail messages [12]. The '80-20 rule' (80% of the time people are use 20% of the items) suggests that designers should ensure that the most frequent items are efficiently accessible in the interface [4]. Researchers have also carried out studies to investigate people's revisitation and re-finding strategies for e-mail [12], web pages [5], and search results [38].

Information on previous interaction can be used to tailor the presentation of items in a visual workspace. For example, in a paged interface, items could be placed across pages in the order of their visit count by previous users, and in a spatial interface, the most popular items could be placed in prominent positions (e.g., top left or central). Alternatively, in a spatial interface, the most popular items could be shown larger than others, or with some other form of highlighting applied to them. The idea of showing interaction history as part of the visual representation of data was introduced as "edit wear and read wear" by Hill and colleagues [19] and has been tested in several contexts, such as an interface design that highlights frequent items [13].

### 2.4 Expertise Development and Spatial Interfaces

Users' patterns of revisitation provide opportunities for interface designers to enable faster retrieval of previously visited items. Early attempts at doing so in menu interfaces examined rearranging menu items, placing the most frequently or recently used items closer to the top of the menu to reduce selection pointing distance [31]. However, rearranging items reduces predictability, because the user cannot rely on items remaining in the same location, which can impair learning, require slow visual search, and necessitate double-checking of items, all of which will reduce performance.

To overcome these limitations, many research systems have exploited spatially stable layouts that allow the development of spatial location memory. Examples include: CommandMaps [29], which presents all of an application's commands in a spatially stable 2D matrix; Space-Filling Thumbnails [9], which presents a spatially stable 2D matrix view of page thumbnail for document page navigation; and FastTap [16], which provides a spatially grid of commands for mobile devices. Evaluations of these techniques have all demonstrated rapid performance once item locations are learned, and overall performance that is no worse than state of the art techniques prior to learning target locations.

Spatial memory has been shown to be resilient to transformations such as rotation and resizing [32], and has worked well when used in a real-world interface [30]. However, despite the large amount of research on spatial memory and spatial interfaces, there is little known about how the approach works with large item sets – which is the question that we explore in the study described below.

### 3  STUDY METHODS

The goals of the evaluation were: to assess the performance of spatially-stable presentation during exploration of a large data set with varying degrees of cue information; to compare that performance to a paged approach, both for initial finding and for revisitation; and to investigate participant strategies and difficulties in both interfaces.

### 3.1  Apparatus and Participants

We built a visual workspace system with two datasets of 700 items each (one showing graphics cards and one showing computer cases). Our criteria in choosing datasets were that the set had 700 items each with a visual representation, that the items in each set were visually similar (to avoid any advantage for the spatial technique that shows all items, such as a popout effect), and that the items should have several elements of metadata for filtering.

We obtained our two datasets from an online product catalogue for computer parts.

The system was built with HTML, CSS, and Javascript, and was deployed locally through a Firefox browser running on a Linux PC. The system showed the visual workspace on a 29-inch 4K monitor; the study also used a second monitor to display a picture of the next target, along with varying cue information that could be used to set filters (e.g., manufacturer and main features). Participants controlled the system with an optical mouse and keyboard.

Twenty participants (12 women, 7 men, 1 non-binary, mean age 26.3) were recruited from a local university and given a $10 honorarium. All participants were highly experienced with WIMP systems and web browsers (all > 2 hours / day).

### 3.2  Study Conditions

We developed custom interfaces to compare spatial and paged approaches to finding and revisitation. Both interfaces had navigation capabilities appropriate to the approach; neither UI allowed text search (e.g., Ctrl-F), in keeping with the exploratory nature of the domain as described above.

#### 3.2.1  Spatial interface

The Spatial interface showed a grid of images that contained the entire 700-item dataset (Figure 1). Each image was 41×26 pixels (19×12mm), and no text was shown in the main view. Items were sorted by price in row-major order, and were spatially stable (items never moved). Filters were shown with highlighting and dimming (all items matching the filters were outlined in green, and all non-matching items were dimmed; see Figure 1).

Mousing over an item in the grid showed a larger image using a pop-up animation. Users could click on an item to inspect it (i.e., show a full-size image and all textual information in the detail panel at top right of the screen, see Figure 2). The currently selected item was outlined in pink in the overview, and items that were recently inspected were outlined in blue.

#### 3.2.2  Paged interface

The Paged interface (Figure 2) organized items into pages with 24 items per page, sorted by price. Each item showed a picture (392×312 pixels, 106×82mm) and summary textual information. Whenever filters were applied, the system organized the filtered subset into a new set of pages (still ordered by price).

Users could navigate the pages in three ways: first, by scrolling with the scroll wheel to see the 12 items that were beneath the initial view in the current page; second, by clicking "Previous" and "Next" buttons to move between pages; third, by entering a number in a "Jump to Page" field.

#### 3.2.3  Amount of Cue Information

In each trial, a separate monitor displayed a picture of the target and a set of cue information that could be used to set filters. We used three levels of cue information: *full* provided the picture and all of the information that was in the item's detail page; *partial* provided the picture and the manufacturer's name; and *minimal* provided only the picture. Examples of these three levels are shown in Figure 3. Our goal in changing the amount of cue information was to serve as a proxy for the varying degrees of recollection that a user would have for the details of previously visited items during an exploration task.

#### 3.2.4  Filter panel (available in all interface conditions)

A filter panel was shown at the left side of the workspace, with categories that were extracted from the dataset (see Figures 1 and 2). For the graphics-card dataset, filters included Type, Memory,

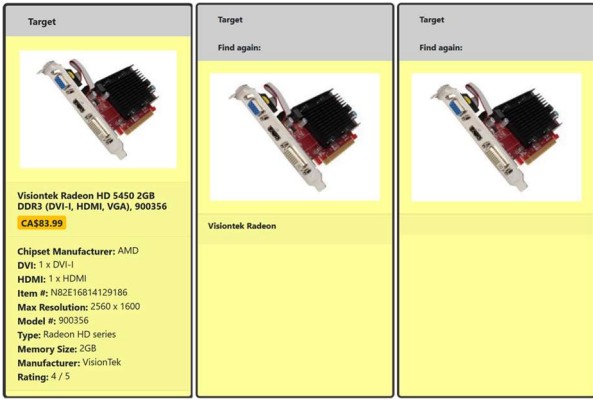

Figure 1: Example of full, partial, and minimal cue information for the graphics-card dataset.

Monitor Interface, Manufacturer, Price, and Rating. For the computer-case dataset, filters included Motherboard Support, Manufacturer, Price, and Rating. (Although there were different numbers of filters for the two datasets, the sets were counterbalanced with interface conditions in the study).

### 3.3 Procedure and Task

Participants were given a brief introduction to the study, and then completed an informed consent form and a demographics questionnaire. They were then placed in an order condition to determine which interface and dataset they would use first (both counterbalanced). Participants were shown a demo of the system, including selecting an item, applying filters, and navigating the interface; they then completed four practice trials with a third dataset (showing memory sticks – used only for training).

Participants then carried out the main part of the study, in which they looked for six different targets over eight blocks of trials. The target was shown on a separate monitor (picture and cue information), and as soon as the target appeared, the user's task was to locate the target in the system, click on it, and press the spacebar to indicate that they were finished. Participants could use any of the navigation features available to them for that condition. Trials were limited to 90 seconds: at this time, the trial completed automatically and the system moved to the next trial.

Blocks showed the six targets in random order (sampling without replacement). Blocks had different levels of cue information, in the following order: 1:Full, 2:Partial, 3:Full, 4:Partial, 5:Partial, 6:Minimal, 7:Minimal, 8:Minimal. Cue information gradually decreased over the blocks, to simulate a reduction in reliance on external materials as a work session continues.

After completing eight blocks with one interface, participants completed a TLX-style perceived-effort questionnaire. They then carried out blocks of trials with the other interface and dataset, finishing again with the effort questionnaire. At the end of the session, participants answered questions about which interface they felt was fastest, most accurate, and best supported remembering, as well as which interface they preferred overall. The study took approximately 60 minutes in total.

### 3.4 Design

The within-participants study used a repeated-measures factorial design, with factors *Interface* (Paged or Spatial) and *Block* (1-8); the eight blocks covered three different *Cue Levels* (Full, Partial, or Minimal) – see Procedure section above for details. Interface order was counterbalanced, with half the participants seeing Spatial then Paged, and half seeing Paged then Spatial. Two different 700-item

datasets were used (graphics cards and computer cases), and these were also counterbalanced with Interface.

We collected several dependent measures for each trial:

- Completion time (from appearance of a cue to pressing the space bar with the target selected, 90-sec. time limit);
- Errors (space bar pressed with the wrong item selected),
- Filter use (number of total actions in the filter panel),
- Navigation actions (number of page changes and scroll events, which applied only to the Paged interface);
- Inspections (number of items clicked on to show details).

We considered three primary research questions:

- *RQ1: which interface better supports revisitation?* Previous work suggests that the spatial interface will lead to faster retrieval as participants become familiar with item locations; however, familiarity may also benefit the paged interface (e.g., participants may memorize the procedure needed to filter and retrieve each target item).
- *RQ2: which interface has faster initial retrieval?* Both interfaces provide the same filtering behavior, but the presentation of the filtered subset is very different. The additional navigation required in the Paged interface could slow it down, but users could find it more difficult to interpret the filters in the Spatial display (where items are not removed by filters, just lowlighted).
- *RQ3: which interface will users prefer, and which will they see as requiring less effort?* The potential advantages of the spatial approach come at the cost of using smaller images and placing all items on the screen at once, which may overwhelm users. Participants may prefer the interface they have experience with (Paged).

We also investigated several other issues as follow-up or exploratory hypotheses, including differences between the interfaces in terms of the number of filters used, the number of errors, and the number of inspections and navigation actions. These were used to investigate reasons for differences in our primary research questions. Each participant carried out 96 trials (6 targets * 8 blocks * 2 interfaces), and with 20 participants, there were 1920 datapoints collected in total.

## 4 RESULTS

No data were removed due to outlying values, but 46 trials were capped at the 90-second limit for Paged, and 56 for Spatial. We analysed our main research questions regarding performance, filter use, inspections, and navigation actions with repeated-measures ANOVA (for factors Interface and Block). Because Cue Level was combined with Block, we consider these factors together when interpreting interactions. Perceived effort and preference data were analysed using Wilcoxon and $\chi^2$ tests. We report effect sizes for significant RM-ANOVA results as general eta-squared $\eta^2$ (considering .01 small, .06 medium, and >.14 large [10]).

### 4.1 Effects of Interface on Retrieval Time

We compared the mean completion time for the two interfaces across trial blocks (as described above, blocks were combined with cue level). A 2×8 RM-ANOVA showed no main effect of Interface on completion time ($F_{1,19}=0.16$, p=0.69), but did show a strong effect of Block ($F_{7,133}=42.8$, p<.0001, $\eta^2=0.30$) and an interaction between Interface and Block ($F_{7,133}=2.39$, p=0.024, $\eta^2=0.03$). As seen in Figure 4, the interaction is apparent in the higher retrieval times for Spatial in the first two blocks (with full and partial cues), and lower times in the last three blocks (with minimal cues).

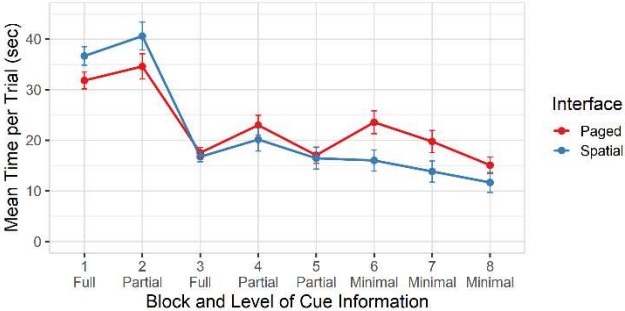

Figure 4: Completion time per trial, by block and cue level (error bars show ± 1 s.e.).

Cue Level and Block were combined in our analysis, so the interaction could arise either because of the degree of familiarity built up across blocks, or because of the different amount of cue information in blocks 1-2 versus blocks 6-8. As discussed below, both factors likely played a role – by the final blocks, participants had built up spatial memory in the Spatial condition that allowed them to overcome the reduced cue information, whereas the Paged interface remained reliant on filtering, so the reduction in cue information caused more problems for participants.

Follow-up Holm-corrected t-tests at each block showed that Spatial was significantly faster than Paged at block 6 (p=.015); no other blocks showed significant differences, although two were close (blocks 1 and 7 at p=.054).

### 4.2 Finding Items for the First Time

There was no statistical difference between Paged and Spatial for the first block of trials (where participants were seeing each item for the first time). However, the test was close (p=.054), and the mean time for Paged (31.8sec) was lower than for Spatial (36.6sec).

To further understand the differences between the interfaces for initial search, we looked at the individual trials of Block 1 (which represent the first time each of the six targets was found). Figure 5 shows that participants in both conditions improved over the six targets. Participants in the Paged interface were faster for trials 2 and 3, but this may be a reflection of the overall level of familiarity that participants would have with this style of interface (whereas the Spatial approach would be new to most). By the end of the block, there is little to separate the two conditions.

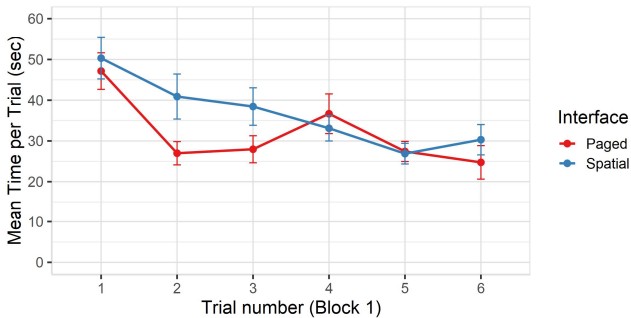

Figure 5: Mean time (± 1 s.e.) for each trial in block 1 (representing the first time each of the six targets was found).

### 4.3 Effects of Interface on Filter Actions

Filter actions were counted as the total number of times the filter criteria checkboxes were clicked, plus the number of times the "Apply" button was clicked. We analysed both the number of filter actions performed in each trial, as well as the total number of filters applied at the trial's completion.

A 2×8 RM-ANOVA showed main effects of both Interface ($F_{1,19}$=27.4, p<.0001, $\eta^2$=0.17) and Block ($F_{7,133}$=8.84, p<.0001, $\eta^2$=0.090) on filter actions. There was also an interaction between Interface and Block ($F_{7,133}$=5.87, p=<.0001, $\eta^2$=0.079). The interaction is evident in Figure 6: the two interfaces have approximately the same number of filter actions in the first block, but while the number remains high for Paged, filter actions for Spatial drops consistently.

The chart shows that each time the amount of cue information was reduced (blocks 2, 4, and 6), filter use rose for the Paged interface (but for Spatial, the only rise was seen at block 6, and this was small). When a block had the same amount of cue information as the previous block (i.e., blocks 4, 7, and 8), filter use decreased in both conditions (although by larger amounts for Paged). These data show that participants using Paged were more reliant on filters overall, and that their filter use was as high in block 7 (with minimal cue information) as it was in block 1 (with full information).

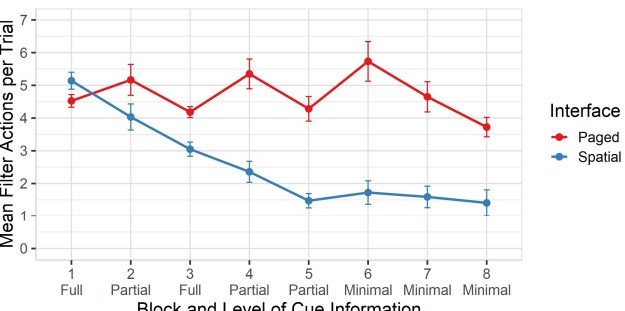

Figure 6: Filter actions per trial, by block and cue level (error bars show ± 1 s.e.).

Post-hoc Holm-corrected t-tests at each block showed that the difference in number of filter actions was strongly significant from block 3 onwards (all p<.0001).

We also analysed the total number of filters in use at the completion of each trial (Figure 7). A 2×8 RM-ANOVA showed main effects of both Interface ($F_{1,19}$=16.8, p<.0001, $\eta^2$=0.15) and Block ($F_{7,133}$=37.4, p<.0001, $\eta^2$=0.29) on total filters. There was also an interaction between Interface and Block ($F_{7,133}$=12.9, p=<.0001, $\eta^2$=0.10). The interaction is shown in Figure 7: as blocks progress, filter use in the Paged interface remains at about two filters per trial, whereas in the Spatial interface, use drops to 0.4 per trial by block 8.

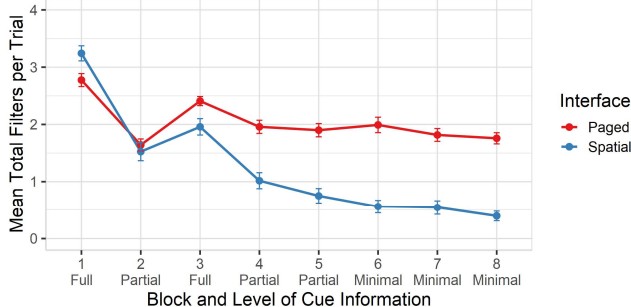

Figure 7: Total filters per trial (± 1 s.e.), by block and cue level.

Post-hoc Holm-corrected t-tests showed that in block 1, more filters were used with the Spatial interface (p=.008), but the reverse was true from block 3 onward, with significantly more filters used in the Paged interface (all p<.0001).

## 4.4 Errors

Errors were counted each time the participant pressed the space bar with the incorrect target selected. We expected that there would be a higher number of errors for the Spatial condition both because the images were smaller in the main view and because people may have guessed at the target as they started to use memory.

Errors were higher overall for Spatial (mean 0.25 errors per trial, s.d. 1.13) than for Paged (mean 0.14, s.d. 0.47). However, a 2×8 RM-ANOVA showed no effect of Interface ($F_{1,19}$=1.70, p=.21), and no interaction between Interface and Block ($F_{7,133}$=1.66, p=.124). There was a main effect of Block ($F_{7,133}$=3.34, p=.0025, $\eta^2$=0.06).

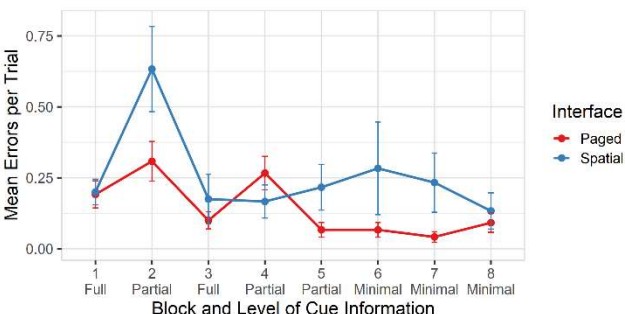

Figure 8: Mean errors per trial (pressing space with the wrong target selected), by block and cue level (error bars ± 1 s.e.).

## 4.5 Inspections

Inspections were recorded whenever a participant clicked on an item in the main view (which provided a larger view in the top-right panel, see Figures 2 and 3). We expected the number of inspections to be higher in the Spatial condition, where the main view provides a smaller image and less textual information; there was less need to inspect items in the Paged interface because there was often enough information available in the main view.

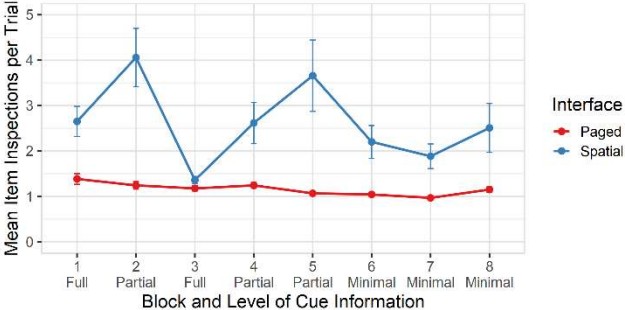

Figure 9: Item inspections per trial, by block and cue level (error bars show ± 1 s.e.). All completed trials required at least one inspection (to select the target).

As expected, RM-ANOVA showed main effects of both Interface ($F_{1,19}$=7.96, p=.011, $\eta^2$=0.086) and Block ($F_{7,133}$=2.25, p=.034, $\eta^2$=0.032) on number of inspections. There was no interaction between Interface and Block ($F_{7,133}$=2.01, p=.058).

## 4.6 Navigation Actions

Navigation actions only exist for the Paged condition, since there was no scrolling or paging in Spatial. We counted page-forward, page-backward, jump-to-page, and scrolling actions (we coalesced scroll events until there was at least one second between events). This measure indicates whether people were building up expertise in the Paged condition (i.e., fewer navigation actions could mean that people are becoming more familiar with the dataset).

Because navigation actions for Spatial are zero, we do not carry out a statistical analysis. As seen in Figure 11, navigation actions mirror filter use. This further suggests that participants in the Paged condition remained reliant on filters, and if the filters could not be set accurately to reduce the subset to a small number, more navigation was required to search through the larger set of items.

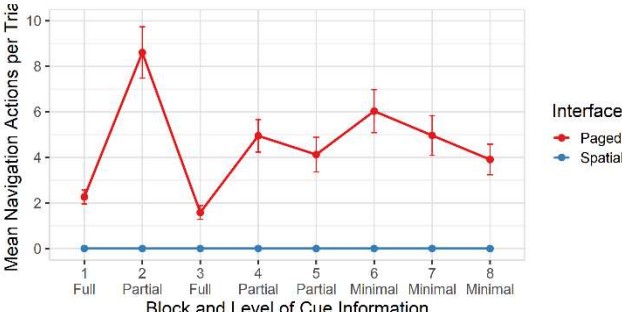

Figure 10: Navigation actions per trial, by block and cue level (error bars show ± 1 s.e.). Note that Spatial has no navigation.

## 4.7 Individual Differences

There were large individual differences, with more than half the participants showing a strong bias towards one or the other of the interfaces. Seven participants were substantially faster (completing tasks in roughly half the time) with the Spatial interface, and five participants were substantially faster with Paged. We return to this issue in the discussion.

## 4.8 Subjective Effort Ratings

Median responses to the TLX-style questionnaire (Figure 11) showed similar ratings on all questions. The only differences were that Spatial was rated slightly higher both in terms of perceived success and frustration. None of the elements were significantly different (Wilcoxon tests, all p>.1).

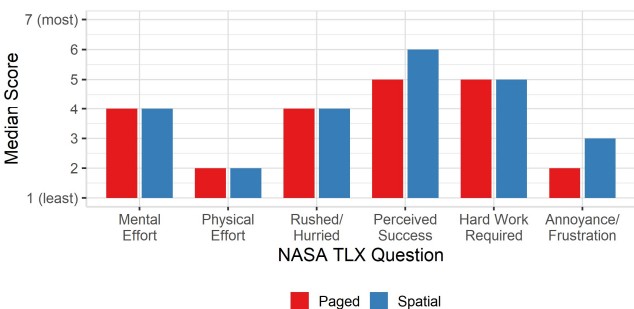

Figure 11: Median TLX responses, by question and interface.

## 4.9 Preferences and Participant Comments

At the end of the study, participants stated which interface they felt was fastest, which was most accurate, which assisted more with remembering items, and their overall preference (Table 1).

Table 1. Participant preferences

| With which interface were you… | Paged | Spatial |
|---|---|---|
| Fastest? | 8 | 12 |
| Most accurate? | 10 | 10 |
| Better able to remember items? | 9 | 11 |
| Which did you prefer overall? | 13 | 7 |

Participants were evenly divided: more thought that Spatial was faster and better supported remembering, and people were evenly split on which was more accurate. Overall preference was in favour

of Paged, with 13 of 20 preferring this interface (Table 1). Chi-squared tests showed no differences (all p>0.05).

Participant comments provide insight into these preferences and ratings. Several participants stated that retrieval with Spatial was faster because they could use memory (e.g., one person said "I found it easier to remember the locations of the items"; another stated "no matter what filters, location never changed"; and a third said "[I] could locate position by memory quickly"). However, one participant also doubted their memory, saying "I don't trust my memory enough." Other participants noted the reduced navigation in the Spatial condition: one participant said "Didn't have to scroll through pages"; and another said "all the items were visible in one screen and [I] did not have to go to different pages." One participant also mentioned that items in the Spatial display could be used as landmarks, stating that finding an item was fast "partly because I could remember what's near it."

Participants who preferred the Paged interface commented on its familiarity (e.g., it was "more similar to online shopping") and the smaller number of items to look at (e.g., "there were less items on the page to choose from"; "it eliminated photos that were irrelevant to the filter"), which also meant that images were larger (e.g., "I felt like I was more sure of my choice because it was a larger image"; "it is less tiring than trying to find in [the] smaller pictures"). Some participants stated that they were better able to remember items with Paged: one stated "I could remember what pages things were on"; and another said "I just needed to remember one or two details about the item, and then I could quickly narrow it down […] so I could get only one page of results."

## 5 DISCUSSION

Our study provided five main findings:

- Initial finding was slower for the Spatial interface, largely due to differences in trials two and three;
- Retrieval time when revisiting items in the final three blocks (where the only cue was the picture) was faster for Spatial;
- Filter use was substantially lower for the Spatial interface, particularly in later blocks;
- Errors were higher for the Spatial condition (although the difference was not significant);
- Perceptions of effort were about the same for the two interfaces; opinions about speed and memory support slightly favoured Spatial, but 13 of 20 preferred Paged.

Here we consider reasons for these results, how our findings can be applied to the design of real-world visual workspaces, and limitations and opportunities for further research.

### 5.1 Explanations for Results

#### 5.1.1 Why was Spatial faster when re-finding?

Trial completion time was faster with Spatial in blocks 6-8. In these later blocks the amount of cue information was minimal, meaning that filters would be less accurate (unless a participant had memorized the cue information from earlier trials) and therefore would result in more items to search through. The stable grid of the Spatial interface provided a remedy for this problem – and as blocks progressed, users of the Spatial interface were able to build up memory of where things were in the grid. Our results show that Spatial users relied less on filters in later blocks and more on their memory of the item's location; and by the final blocks, participants were using no filters at all in more than half the trials.

As discussed earlier, having minimal cue information to use in filters is a common occurrence in many exploration tasks – users may remember details about an item (e.g., "the case with red LEDs") that are not part of the filter capabilities.

One main difference that affects performance in these situations is the type of memory that users can develop with the two interfaces. With Spatial, users build location memory for where items are in the grid; in contrast, users of the Paged interface must remember what page the item was on, and/or what combination of filters reduced the set enough to make the item easy to find. Although participant comments indicated that people were able to remember things in both conditions, our results suggest that for most participants, developing spatial memory was easier than remembering either pages or filter combinations.

#### 5.1.2 Why was Spatial slower with initial finding?

Trials with the Spatial interface were slower in block 1, and also used more filters (although only the filter difference was significant at p<.05). Finding things for the first time is an important part of any visual workspace, so these differences must be considered further. Figure 6 (completion time for each trial in block 1) shows that only two of the six trials were substantially different. We checked the individual trials for filter use and inspections, but these data did not account for the differences in trials 2 and 3. It may be that the differences result from participants' familiarity with Paged interfaces: that is, people may have been comfortable with the Paged interface right away, whereas it took them a few extra trials to get used to working with the Spatial interface. This is an issue to explore further, but it is important to note that if general familiarity is the reason for the difference, this will only happen once (not every time a new target is introduced). In future work, we can assess this question by giving participants a new set of targets at the end of the study (when people are familiar with both interfaces).

#### 5.1.3 Why were participants' preferences split?

Participants' comments showed that they understood the strengths and weaknesses of both interfaces – for example, people noted the advantages of remembering items with Spatial (and even remarked on the fact that nothing moves in the layout), but also commented on its drawbacks in terms of smaller images and reduced focus on the filtered subset. Similarly, participants clearly recognized the increased visibility of items in the Paged interface, as well as its more familiar style; but also recognized the additional navigation that was required. The division in opinions and preferences mirrors the strong individual differences seen in the performance data.

It seems clear that the Spatial interface is subjectively harder initially, but improves for most users with continued use because it provides the opportunity to switch to a faster retrieval mechanism as the user becomes familiar with a dataset. The evenly-divided opinions and preferences mean that designers should consider incorporating both approaches rather than choosing only one.

### 5.2 Generalizing the Findings to Real-World Systems

Here we consider how the two approaches tested in our study can be combined with other features (and what this will mean for performance), the scale limit of the spatial approach, and whether a hybrid interface could get the best of both UIs.

#### 5.2.1 What will happen with the addition of recency caches, favourites lists, history lists, and read wear?

As introduced earlier, real-world systems have recognized the difficulty that users have in getting back to previously-visited items, and have incorporated several techniques to try and address the problem. All of these techniques can assist the user in certain cases, but all have limitations as described earlier. Although our intention in the study was to compare the performance of the two approaches independently (rather than actual implementations), it is important to consider how the interfaces will perform once these additional support techniques have been added.

*Recency caches.* "Recently viewed" mechanisms are common, but are ineffective if the user looks at more items than the cache size before wanting to revisit something. If the user's working set

is smaller than the cache size, the technique will work well. However, showing recent items in a separate list (as is often done) could be detrimental for a spatial approach, because the separate presentation does not allow the user to make use of their spatial memory, and does not help to reinforce that memory. A likely better strategy is to visualize recently-viewed items with a read-wear technique (described below).

*Favourites lists.* If the user explicitly marks any items that they want to revisit, then a separate display showing only these items could solve many of the problems of revisitation. This approach can work well with either the Paged interface (as a separate display) or the Spatial interface (as a mark on the item's image in the grid view). However, a well-recognized problem with this technique is that users often do not remember to flag items, or do not know when looking at an item that they will want to revisit it. A favourites list that only shows some of the important items in the collection can be frustrating for the user.

*History lists.* Showing a user's interaction history can solve the problem of a small recency list and can also help when the user does not remember to flag an item of interest. The drawbacks of a history list are that it can itself be large if the user has visited many items – finding an item in a large history list can be almost as difficult as finding an item in the entire collection. Again, the efficacy of this technique depends on the task (e.g., whether the user browses a large number of items or focuses on a few).

*Edit wear and Read wear.* Visualization of the user's interaction can show recency, frequency, and history – and the idea can be extended to show items that have been explicitly marked (e.g., with a star glyph). Read wear can be applied to both paged and spatial interfaces – in paged systems it can assist the user once they have set their filters (potentially reducing the number of filters that are required), and in spatial systems it can be extremely effective because all of the items are visible at once.

### 5.2.2 Combining the best of both interfaces

A hybrid system could potentially provide a transition to memory-based retrieval while still retaining the strong visibility and focus of the paged approach. One possibility is to provide two representations that are visually linked – e.g., half the screen for a grid overview, and half for a paged view that shows the current filter subset. To help build spatial memory, the current item in the paged display would be highlighted in the overview. The implicit visual indication of where things are in the overview can provide support for the development of location memory – but the main question with such an interface is how to encourage people to switch over to using their memory in the overview display [31].

A second possibility is to increase the size of filtered items in the grid view – either shrinking context items with a fisheye function, or with larger pictures over the non-matching items. Because filtered items will remain close to their original locations, the user can develop at least an approximate memory for item locations.

### 5.2.3 What would happen with Spatial in larger collections?

Our system used datasets of 700 items on a 29-inch display, meaning that each image in the spatial grid was 19×12mm; our experience suggests that this was close to the limit of what would be acceptable to users. It may be possible for larger collections to be adequately displayed using larger screens (e.g., [4,42]) – a larger display will improve visibility of each item, but it is possible that larger grids will make the interface more overwhelming for users. Larger screens do raise the possibility of different layout techniques such as the "image plot" (a scatterplot with images for each item [40]) which could improve performance because groups of items create additional spatial landmarks, and because different variables can be represented on the X and Y axes).

When collections are much larger (thousands of items), using a grid of images sorted by one variable is unlikely to work well due to space limitations. However, there are many tasks where datasets are similar to or smaller than the 700-item sets we studied, and in these situations the spatial approach is a viable design possibility.

### 5.2.4 What do the large individual differences imply?

Paged interfaces should not be abandoned, as they cater to some people's preferences. However, several participants in the study performed poorly with Paged throughout the study, but were successful with Spatial; this split performance should inform design decisions. Further, there are many examples where systems provide additional interfaces for experts (e.g., hotkeys, gestures, or techniques such as Maya's 'hotbox' [24]). Providing a spatial interface may improve usability for users who prefer the spatial interface, and promotes expertise development for a given dataset.

## 5.3 Limitations and Future Research Directions

Our study was limited by several experimental constraints, and many of these can be addressed by future research:

- Our retrieval tasks were artificial (to improve control), and so further studies are needed with more realistic exploration and decision-making tasks. In addition, as discussed above, we did not study the interfaces in combination with other support techniques (e.g., recency lists and read wear); future work will examine systems that include these features.
- Our cue information always provided an exact picture of the target, but in real-world use, people will have varying degrees of memory for an item's image and characteristics. Spatial interfaces may perform better in these circumstances (as spatial memory can stand in for attribute memory), but further studies are needed to examine these effects.
- We examined immediate spatial memory, but did not test retention over longer time periods. Although many tasks in visual content collections are short-term, further studies should explore whether spatial memory persists (as is suggested by previous work [27,28]).
- Spatial abilities are known to vary widely, and in future studies it will be interesting to see if innate ability (e.g., using a prior object-location memory task [11]) predicts a participant's success with the spatial interface.
- As discussed above, future studies should introduce new targets after participants are experienced with the spatial interface, to determine if the initial differences between the interfaces are due to inexperience with the approach.
- Our spatial interface did not add "artificial landmarks" to the grid; these are known to assist the development of location memory [38], and future work should consider their effect in work with visual content collections.

## 6 CONCLUSION

Finding and revisitation are common tasks in large visual workspaces, and filters are often used to assist users as they look for items. Paged presentation of filter results can make it more difficult for users to remember the locations of important items; spatially-stable presentations do not have this problem, but there is little known about whether the spatial approach will work with larger collections. We carried out a study comparing paged and spatial designs using a 700-item dataset (which is much larger than previous studies of spatial interfaces). We found that although initial finding was slightly slower, revisitation with the spatial interface was faster and required fewer filters as participants gained experience. Our results add to knowledge about spatially-stable interfaces, both in terms of their limitations for initial finding and overall preference, as well as their ability to support a transition to faster revisitation even with large datasets of hundreds of items.

## ACKNOWLEDGMENTS

This research was supported by the Natural Sciences and Engineering Research Council of Canada, and the Plant Phenotyping and Imaging Research Centre.

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
