# OpenReview forum: "Testing the Limits of the Spatial Approach:  Comparing Retrieval and Revisitation Performance of Spatial and Paged Data Organizations for Large Item Sets"
_graphicsinterface.org/Graphics_Interface/2020/Conference — GI 2020_

### Official Review · AnonReviewer2 · 2020-02-05
**Well designed experiment, however focus is unclear and exposition is too lengthy**

**Rating:** 6
**Confidence:** 4

**Review:**

This paper reports on the results of an experiment that evaluated the influence of two layout techniques on retrieval and revisitation activities when working with large visual datasets. The outcome of the study was that the spatially-based interface lead to faster revisitation activity and the use of fewer filters compared to a paginated interface. The study also found that the spatial interface also led participants to be slower when initially attempting to find content in the dataset.

I thought that this was an interesting paper to read. I found the focus on a more “classical” HCI problem to be refreshing and well motivated. The experimental methodology was also quite straightforward (i.e., a clear and simple design was used) and aside from issues with the focus of the metrics (mentioned below), the paper sets up a nice comparison between the two interfaces. There are some concerns that I have with the paper, however.

First, the contribution of this research is the finding that participants used fewer filters and were faster as use with an interface increased. However, the paper presents three research questions and at least seven different metrics to these questions to arrive at the above conclusion. Because the research questions are not referred to again in the latter half of the paper, there is no justification for why these research questions or this number of metrics were used, some of the findings were not significant or only significant for the Block factor or were significant but had a small effect size, and there were no corrections applied to account for the number of metrics that were evaluated, it is difficult to understand if the research questions actually informed the comparison and subsequently why 10 pages are necessary to report on these results. There needs to be a clear focus and mapping between research questions and metrics so that the reader can find the contribution within the presented data and not get weighed down in data that ultimately doesn’t add to the overall story of the paper.

In addition to the focus challenges, I also found that the paper was too long and could benefit from a condensing pass to make it better fit the depth of the contribution (which I see as about 6 or 7 pages excluding references). In the introduction, for example, many parts can be trimmed or removed without losing clarity (e.g., the second paragraph, the explanation of the experiment, and all of the Figures). The findings are actually best summarized in the single sentence in the conclusion so the contribution statement in the Introduction can probably be condensed into one sentence as well. Other sections have a lot of repetition (e.g., Design first paragraph) and circular self-referencing (e.g., referring forward to a forthcoming section or backwards to a previous section). Given the main finding of the paper, the discussion and limitations are too long (close to 2 pages) and I don’t see the relevance of devoting a whole page to the ‘Generalizing the Findings to Real-World Systems’ section.

Lastly, there are a number of claims, decisions, and terms made in the paper that are unclear. For example, why were 700 items chosen for each dataset and how representative is this of “large” datasets (a reference would be beneficial), how was 19 mm x 12 mm “close to the limit of what would be acceptable to users”, how “participants using Paged were more reliant on filters overall” given that this did not occur on Block 1 as per Figure 7, what are “location memory” and “lowlighting”, and where does Figure 1 come from? While the experiment is well explained, it also seems that a time limit was imposed on trials after the experiment was completed. It is unclear how this time limit influenced the retrieval time metric given that means and standard errors are given (i.e., were these trials removed from the dataset or were any values over 90 seconds scaled down to 90 seconds).

Although I appreciate the overall goal of this paper, the paper is too long given the contribution and the important results are hidden amongst too many unnecessary metrics, exposition, and discussion. If others are strongly in favor of accepting this paper then  would recommend that it be shepherded (if possible) so that it can be condensed and better focused.

Other:
-	Figure 3 refers to “in this trial” but the figure and caption do not refer to trials but rather an interface
-	Page 4: “little is known about how the approach works with large item sets” spatial memory is not an approach
-	Page 5: “previous work suggests that the spatial interface will lead to faster retrieval …. “ citation needed
-	The appropriate scientific symbols should be used for mean and std
-	It is common to first present all of the statistical results and then provide some analysis of them in the text rather than present the main effects and interactions, synthesis, and then the post-hoc analysis. Dividing the main effects and interactions from  the post-hoc tests makes it difficult for the reader to understand the synthesis given that in many cases the post-hoc tests explain the interactions and thus synthesis that is presented.
-	Figure 12 is missing an error measure
-	There is quite a bit of use of bulleted lists in the text. These should be converted into sentences to allow for easier readability and better organization of the themes / findings (Page 2, 5, 8, and 10).
-	An inconsistent citation format is used in the reference list

---

### Official Review · AnonReviewer1 · 2020-02-11
**Solid but too long**

**Rating:** 7
**Confidence:** 4

**Review:**

The paper "Testing the Limits of the Spatial Approach" presents a study comparing a spatial technique for finding and revisiting objects in a visual content collection with a standard paginated layout common on the web. The tasks in the study involve searching in 700 graphics cards and computer cases, filtering on various attributes such as memory size, etc.
Results show that the spatial layout performs better for refinding items and filtering was more reliant, but the traditional paginated layout outperforms on the initial search.

The paper presents a well designed and well-executed study with a good and nuanced analysis of the results. It is clear that the spatially-stable technique has potential, but that it is not purely black-and-white, which the authors carefully detail. The paper does not present groundbreaking research, but it is a solid piece of work providing evidence that it is worthwhile exploring the use of spatially-stable organization techniques for certain search and filtering tasks.

While the paper is generally well-written I find it to be too long. The introduction and related work could (and should) be shortened to around two pages. I also found the discussion overly verbose and I would suggest shortening it as well to make the paper more on point.

I missed a better description of the tasks, it was not clear to me what the participants were supposed to find, and why and under what conditions they needed to refind items. Also, do the tasks reflect typically search and filter tasks (e.g., on the web)?

Overall, I find the results of the paper to be interesting, and the study well designed and executed. I missed a bit of clarity around the tasks, and I would strongly recommend the authors to shorten the paper significantly.

---

### Official Review · AnonReviewer3 · 2020-02-11
**Limited in contribution, but extends prior research on spatially-stable layout and offers future directions for research.**

**Rating:** 6
**Confidence:** 3

**Review:**

This work investigates the effectiveness of a spatially-stable layout for finding items within a large item set (700). The authors conduct a within-subject study (n=20) to compare paged and spatial layout performance with two datasets. The results show that while spatially-stable layouts are faster for revisitation (from spatial cognition), they were less accurate. The overall performance times were similar across both types of interfaces.

While the contribution of this work is somewhat limited, the main finding that end-users are able to leverage spatial memory for larger datasets both supports and extends prior work (e.g., [9] evaluates thumbnails of text documents with 300 pages). The specific findings presented in the paper (interface, navigation, finding items for the first time) offer useful insights for future research on spatial-layouts.

Pros:
The paper is well written and the main arguments are easy to follow
Related work is quite comprehensive, but can be synthesized better with a stronger connection to this work.
The study design is sound and follows the protocol from prior research (repeated-measure factorial design)
The analysis is also clear and the authors do a good job of summarizing the findings.
The discussion does a good job of expanding on the findings, and the generalization section offers directions for future research. (Although more discussion about (1) the advantages and tradeoffs between spatial memory and remembering "attributes"  about items, and (2) time vs. accuracy tradeoffs, and (3) dataset characteristics would have been helpful.)

Comments and clarifications:
I had a set of comments that can potentially be addressed in the final version of the paper:
Besides accounting for popout effects, were there other reasons for selecting the datasets used in the study? Did participants have familiarity/ domain understanding of the datasets? Would familiarity change the current findings?
I was a bit unclear about how were the blocks*targets presented, in what order..block1-target1, block1-target 2... block8-target 1...  or was it random?
What was the spatial distribution of the targets selected in the study? How far apart were they?
The text description labels the cues as full, partial, and minimal which is inconsistent with the axis labels in the figures: All, medium, least.
There is no discussion about visual cues on recently selected items (blue outline) and recency list on participants' performance. Same for the use of popup animation to see a larger image in the spatial view.

---

### Meta-Review · Area_Chair1 · 2020-02-12

**Recommendation:** Accept
**Confidence:** 4

**Metareview:**

All reviewers agree that the paper presents some interesting findings, although they are somewhat limited in scope. They all find that the experimental methodology is sound and that the paper is well written.
The reviewers agree that the paper in its current state is too long and there are a number of sections that can be condensed. Reviewer 2 has concrete suggestions for how to shorten then paper.
There was some unclarity in regards to the tasks, and to how blocks*targets were presented that needs to be addressed in a final version. Also, reviewer 2 points out that the authors do not refer back to the research questions in the latter half of the paper.

Overall, I lean towards accepting the paper. However, the authors should carefully consider the suggestions made by the reviewers, particularly in regard to shortening the paper and clarifying the questions they have raised.

---

### Decision · Program_Chairs · 2020-02-18

Accept